# Study on the Mechanical Properties of Monocrystalline Germanium Crystal Planes Based on Molecular Dynamics

**DOI:** 10.3390/mi13030441

**Published:** 2022-03-15

**Authors:** Linsen Song, Juncheng Song, Junye Li, Tiancheng Wang, Zhenguo Zhao

**Affiliations:** 1Ministry of Education Key Laboratory for Cross-Scale Micro and Nano Manufacturing, Changchun University of Science and Technology, Changchun 130022, China; sls20221@163.com (L.S.); sjc20221@163.com (J.S.); wtc202201@163.com (T.W.); zjg202201@163.com (Z.Z.); 2Chongqing Research Institute, Changchun University of Science and Technology, Chongqing 401135, China

**Keywords:** molecular dynamics, monocrystalline germanium, nanoindentation, mechanical properties

## Abstract

Nanoindentation and atomistic molecular dynamics simulations of the loading surface of monocrystalline germanium were used to investigate the evolution of the key structure, the force model, the temperature, the potential, and the deformable layer thickness. The mechanical characteristics of typical crystal planes (001), (110), and (111) of the crystal system were compared under load. It was observed that the hardness and stiffness of the (110) plane were greatest among the three crystal planes, whereas the hardness and stiffness of the (111) plane were lowest. Moreover, the deformation layers at the ends of both planes were basically flat. The processing efficiency of the (111) surface was higher; thus, the (111) surface was considered the best loading surface. It was concluded that the subsurface defects of the monocrystalline germanium (111) plane were smaller and the work efficiency was higher during the processing of monocrystalline germanium, making it ideal for monocrystalline germanium ultra-precision processing.

## 1. Introduction

Germanium finds its applications in the fields of infrared optics, missile guidance systems, night vision goggles, semiconductors, thermal imaging systems, high-frequency electronics, etc. It becomes difficult to obtain a perfect optical surface. The mechanism of material removal and machined surface characteristics play a great role in manufacturing [1]. As a crystalline material of diamond structure, single-crystal germanium has the characteristics of high hardness, high brittleness, and easy collapse and fracture. In optical parts and semiconductors, the polishing accuracy and the distribution of internal defects are important factors that affect the service life. In crystal defects, the light propagation is limited to an increase in the light intensity. Moreover, the lattice distortion potential field will change the scattering path and the mobility carrier [2]. The surface defects and subsurface defects produced during processing must be strictly controlled. Therefore, the determination of the mechanical properties of monocrystalline germanium is a major focus of the study of ultra-precision machining of monocrystalline germanium parts. Because the diamond monocrystalline germanium lattice structure at normal temperature and pressure has a typical anisotropy, there is a need for mechanical property testing of each crystal face to obtain an optimal working surface.

The nanoindentation hardness test method, also known as depth-sensing indentation, is the application of a small volume of sample to test its nanoscale mechanical properties, such as load–displacement curve, hardness, elastic modulus, fracture toughness, viscoelasticity, and creep behavior. Nanoindentation is one of the most commonly used methods to test the mechanical properties of materials. Nanoindentation equipment can record load, stroke, temperature, and other data in real time through high-resolution sensors and high-precision controllers [3].

The basic process involves the indentation test sample with the mechanical properties of a hard material tip (typically, made of a very hard material such as tungsten or diamond) being pressed into, thereby determining unknown properties, experimenter-defined values, or maximum depth. The load of the probe increases continuously from the time the probe touches the sample until it reaches the value defined by the experimenter. At this point, the load can be kept constant over time or directly unloaded.

The most commonly used analytical method for nanoindentation was proposed by Oliver and Pharr in 1992, where samples are loaded with a constant pressure onto a deep probe by binding to poles constructed using the contact area function [4]. In addition, the analysis theories of nanoindentation also include the strain gradient theory [5], Hainsworth method [6], volume density method, and finite element analysis. There are four types of probe indenters commonly used in nanoindentation experiments: spherical probe, conical probe, Vickers probe, and Berkovich probe.

Nanoindentation or ultra-precision machining on a macroscopic scale makes it difficult to observe the inside of the process of the base body, which can only be achieved using an empirical formula, mathematical modeling, etc. Accordingly, a microscale test piece can be observed more intuitively. To investigate the internal situation of the workpiece, researchers typically use finite element analysis, mesoscopic dynamics, molecular dynamics, or other tools.

Wang et al. calculated the hardness of monocrystalline copper by simulating its nanoindentation and analyzed its elastic recovery characteristics [7]. Vardan et al. simulated nickel twin crystals using molecular dynamics and conducted nanoindentation experiments on the strong and weak grain boundaries by twisting the boundary at a certain angle; they obtained different mechanical properties and carried out the experiment according to the observed microstructure [8]. In addition, they also simulated the nanoindentation process of a graphene-enhanced nickel substrate. By placing the graphene sheet at different depths of the nickel plate to test its mechanical properties, they found that the dislocations generated by the indentation near the indenter could not penetrate the graphene sheets. However, they could bend around the sheet to reach a position below it. If the indenter contacted the flake, dislocations nucleated in the nickel layer outside the flake. At this time, interface cracks occurred between nickel and graphene [9].

Zhao et al. simulated cubic boron nitride to observe the characteristics of the crystal at each stage of the indentation, distinguished the types of dislocations generated during the test, and calculated the energy required for the dislocations in each slip direction [10]. Hui et al. observed the dislocation nucleation phenomenon by changing the number of layers and the number of contact surfaces of twinned copper with high-entropy alloy FeCoCrNi nanolayers, and they obtained nanolayers with high strength and good plasticity [11]. Prasolov et al. performed nanoindentation simulations on a gallium arsenide surface under low-temperature conditions; a stable dimer of arsenic was formed during the surface reconstruction process and would not disappear [12].

The nanoindentation deformation behaviors of Cu 80 Zr 20 (A)/Cu 20 Zr 80 (B) amorphous/amorphous nanolaminates were studied using molecular dynamics (MD) simulation, aiming to investigate the effects of heterogeneous interface and layer thickness on the hardness. It was found that there is a strong length scale dependence for the mechanical properties of amorphous/amorphous nanolaminates [13].

According to a comparison of the nanoindentation behaviors of nanotwinned FeNiCrCoCu HEA (nt-HEA) and single-crystal FeNiCrCoCu HEA (single-HEA), the plastic deformation of nt-HEA was dominant [14].

The deformation characteristics and plasticity mechanisms of a WC–Co composite were demonstrated using molecular dynamics simulations on an atomic scale. It was found that the nucleation, expansion, and interaction of dislocations are distinctly dependent on the orientation where the load is applied [15].

In order to reveal the micro-crack behavior of single-crystal titanium under nanoscale shear stress conditions, a molecular dynamics model of single-crystal titanium was constructed by Li et al. [16].

Wang et al. investigated the crystallographic orientation-influenced indentation size effect in Berkovich nanoindentation tests of single-crystalline copper, using the nonlocal crystal plasticity finite element approach and specifically designed experiments [17].

The onset of plasticity in a single-crystal C60 fullerite was investigated by nanoindentation on the (111) crystallographic plane [18].

Another study reported on the responses of zirconia materials with distinct microstructures to nanoindentation associated with diamond machining using a Berkovich diamond indenter [19].

The mechanical properties of undoped and 2.0, 4.0, and 6.0 mol.% Mg-doped LN single crystals, grown using the Czochralski technique, were investigated using nanoindentation studies to understand the mechanical deformation behavior with an increase in doping [20].

The mechanical response of nonhydrogenated DLC films with different *sp*^3^ concentrations was investigated using nanoindentation experiments, revealing unexpected plastic deformation mechanisms that were not previously considered [21].

Tongfei Tian and coworkers introduced the factors affecting shear-thickening fluid. The composition and rheological properties of the shear-thickening fluid were demonstrated [22].

Cinefra studied the effect of fluid on the hydrodynamic response of nanocomposite tubes. The results showed that the hypothesized nanoparticle agglomeration reduced the velocity and frequency of the critical fluid [23].

In this paper, the nanoindentation test of each loaded surface of single-crystal germanium was simulated by molecular dynamics, and the mechanical properties of typical cubic crystal planes (001), (110), and (111) under loading were compared.

## 2. Preparing for Simulation

### The Establishment of Single-Crystal Germanium Nano-Indentation Molecular Dynamics Model

The lattice type of germanium is a diamond cubic lattice structure with a lattice constant of a = 5.658 Å. The mechanical properties of different monocrystalline germanium crystals can be used to determine the optimal structure by analyzing three typical cubic faces:(100), (110), and (111). Diamond is a cubic crystal structure in which the face-centered cubic structure moves along the body diagonal by one-quarter of the diagonal length and is nested with the original face-centered cubic. The three experimental crystal faces each have four (001) crystal faces. The (110) crystal plane has two symmetry directions, whereas the (111) crystal plane exhibits triple symmetry. The monocrystalline germanium nanoindentation model established in this study is shown in Figure 1.

In Figure 1, the boundary layer is shown in green, the constant temperature layer is shown in light blue, the Newtonian layer is shown in dark blue, and the triangular pyramid probe is shown in red. The cube diamond was cut and rotated through the (111) crystal plane using the atomsk software and then imported into LAMMPS. The distance between the probe and the test piece was 3 Å. The size of the Newtonian layer was 24a×24a×13a, and the constant temperature layer and the boundary layer covered five surfaces with the exception of the loading surface of the Newtonian layer, each with a thickness of 1a. The loading model of each crystal plane was about 100,000 atoms. The experimental model was first relaxed to ensure the stability of the system.

## 3. Simulation Process Analysis

### 3.1. Overall Data Analysis of the Loading Process

In this experiment, the three crystal faces (001), (110), and (111) were loaded vertically to the model. The three coordinate axis directions of the three crystal planes are shown in Figure 2, where (100) is equivalent to (001). In this experiment, the loading depth was set to 3 nm, the loading speed was 10 m/s, and the probe was unloaded at a speed of 50 m/s upward, with a lift height of 2 nm.

First, the overall data of the model were analyzed. The most widely applied analysis method is the load–displacement curve. Figure 3 shows the load–displacement curve of the (001) crystal plane under vertical load. It can be observed that, as the loading progressed, the depth of the probe embedded in the specimen continued to deepen, the force on the probe increased continuously, and there was a force fluctuation with gradually increasing amplitude. The fluctuations represent the continuous transformation and collapse of the lattice. In the nanoindentation molecular dynamics simulation, the material exhibited a stage of elastic deformation, which was specifically expressed as a continuous point with strong linearity at the initial stage of loading, as shown in Figure 4. The area covered by I in the figure can be regarded as a linear phase. 

In order to determine the stiffness of the monocrystalline germanium (001) crystal plane under loading, linear regression was performed on the points from unloading to separation. The results of data processing are shown in Figure 5. The slope of the linear regression was about 872, indicating a stiffness of 872 N/m.

The load–displacement curves of the (110) surface and (111) surface are shown in Figure 6. As shown in Figure 6, the elastic phase was very short when the load was small in the early stage of loading, and the load quickly began to fluctuate, indicating the brittleness of the monocrystalline germanium material. It can be seen from Figure 6 that the load–displacement curves similar to those of the (001) plane in Figure 3. However, the corresponding maximum loads on different surfaces were different. It can be seen that, under the same loading depth, the (111) surface was resolved. It required the least force and the least work. The unloading slopes of the (110) plane and the (111) plane were calculated to obtain the unloading stiffness. The stiffness of the (110) plane was about 968 N/m, while the stiffness of the (111) plane was about 867 N/m; thus, the stiffness values were ranked as follows: (111) < (001) < (110). Figure 7 shows that the maximum compressive force had a high positive correlation with the calculated stiffness.

### 3.2. Atomic Transient Analysis

Next, we analyzed the difference in crystal plane deformation by observing the arrangement of atoms in the model. In the diamond structure, the atomic bond length is $\frac{\sqrt{3}}{4}$ of the lattice constant, i.e., the bond in monocrystalline germanium. The length is about 2.45. In order to be compatible with the change in bond length caused by atomic oscillation, it was set to form a bond when the distance between germanium atoms was within 0.26 nm. The unloaded model was observed from directly above the loaded surface.

In Figure 8, the loaded surface can be observed from directly above the model. Because the atoms have their own vibration during relaxation, which affects their observation, we deleted the observed Newtonian atoms, leaving thin slices. It can be seen from the top views of the three loaded surfaces that, when monocrystalline germanium was viewed from three typical crystal directions, there was an atomic cycle repeatability. When looking down from the crystal plane (001), the meshes in the covalent bond mesh were square, the meshes in the (110) plane were hexagons, and the meshes in the (111) plane were equilateral triangles. The mesh density could be ranked as follows: (111) plane < (001) plane < (110) plane. In relation to the relationship between stiffness and maximum load in Figure 7, we can see that a lower bond density on the crystal plane correlated with increased maximum load and stiffness.

In order to facilitate observation upon slicing the model, the slice thickness was set to twice the lattice constant, i.e., 11.316 Å, and the slice direction was the [100] direction under the coordinate system. The schematic diagram of the slice is shown in Figure 9.

#### 3.2.1. Analysis of the Atomic Transient Diagram Loaded on the (001) Crystal Plane

First, we studied the loading process of monocrystalline germanium with a loading direction of (001) and explored the relationship between the loading depth and the lattice deformation layer by observing the transient image of the atomic slice.

When observing the specimen from the [100] direction, it can be seen that the topology of the specimen in this direction was a square replication continuation with a 45° inclination. As shown in Figure 10, when the stroke reached 0.02 nm, the atomic bond at the bottom of the probe began to break, and the atom closest to the probe was first squeezed away from the nearest atom separated by the tip of the probe, indicating the beginning of the lattice cleavage process. After that, the surface atom situation became complicated. To facilitate observation, we deleted the atoms identified as diamond structure. When the stroke reached 0.04 nm (Figure 11), the coordination number of the atom directly under the probe tip increased to 5. At the same time, under the influence of the plane of the contacting triangular pyramid probe, the three bonds of the regular tetrahedron were almost distributed on the same plane.

The potential energy of the atoms is shown in Figure 12c. Firstly, we can see that the atoms on the surface of the specimen had surface activity; thus, the potential energy was higher. Secondly, we can find that the potential energy of the atoms directly below the probe was significantly lower than that of the surrounding atoms.

When the loading stroke reached 1 nm, some atoms with a distribution number of 0 appeared in the deformed layer. However, when we increased the cutoff radius for identifying the coordination number, the coordination number of these “isolated atoms” increased significantly. A large number of atoms were identified more than 2.6 Å and less than 3.2 Å away from these atoms. Observing the potential energy distribution map, we can find that the lowest potential energy was still concentrated directly under the probe, and the surface potential energy in the loading area was still higher than that in the subsurface area.

Figure 13 shows the coordination number distribution and potential energy distribution when the loading stroke reached 3 nm. In the region where lattice deformation occurred, the potential energy distribution and coordination number distribution had a high degree of consistency with the previous situation, and they were all directly below the probe. Moreover, the potential energy was higher on the contact surface with the probe. The shape of the subsurface phase change layer was a circular arc. Under the loading action of the probe, the surface of the test piece near the loading area of the probe was pulled by the loading part, and part of the surface was depressed, with arcs on both sides.

After the probe was unloaded, the surface of the specimen rebounded, proving that the specimen accumulated part of the elastic potential energy during the loading process as shown in Figure 14. In addition, the *z*-coordinate of the lower surface of the subsurface phase change layer did not change significantly, indicating that the elastic potential energy was basically accumulated from the phase change layer.

Next, we summarize the thickness and width of both sides of the subsurface layer of monocrystalline germanium specimens at different pressure depths. Since there are many atoms undergoing phase change under the influence of loading, this can be used as a unified standard. The *z*-axis distance between the probe tip and the lowest atom whose coordination number changed was recorded as the thickness of the subsurface layer. The statistics are shown in Figure 15.

The line graph between the deformed layer under surface loading and the loading depth revealed that, at the beginning of the loading process, the depth of the deformed layer fluctuated continuously within 0.5 nm. The springback characteristics of the initial specimens were mainly provided by the standard lattice of monocrystalline germanium. When the stroke reached 0.4 nm, the thickness of the deformed layer began to rise steadily, which means that the lattice carrying capacity of the uppermost layer of the specimen reached its limit. In the process of increasing loading depth, the phase change not only existed in the area directly below the probe, but also extended to both sides. From the observation process of the transient atomic state, we found that the lower surface of the subsurface deformation layer constantly switched between a sharp corner shape and a flat shape. Furthermore, the thickness of the deformation layer often decreased at the bottom and both sides. The “sharp corners” of the diamond structure were restored. When the subsequent load became larger, there were more atoms in the normal lattice close to the subsurface deformation layer; thus, each atom was more evenly loaded. Eventually, the lattice could not withstand the pressure directly above it and became a subsurface deformation layer. During the loading process, the thickness of the deformed layer gradually increased, showing a relatively high linearity. In order to gain a deeper understanding of the change trend of the thickness of the deformed layer, a third-order polynomial regression was performed on the points, shown as the red line in Figure 15.

It can be seen from the area covered by the deformable layer in Figure 13 that it was far from reaching the constant temperature layer. This simulation did not cause changes in the thickness of the deformed layer due to the size effect. The highest-order parameter in the regression function of the curve was negative, and there was an obvious trend of slowing down in the later period. Therefore, it can be judged that, when the vertical load continued to increase, the thickness of the final deformation layer no longer changed.

#### 3.2.2. Analysis of the Atomic Transient Diagram Loaded on the (110) Crystal Plane

As observed from the loading surface direction of the model where the (110) plane was the loaded surface, monocrystalline germanium had a clear layered structure from top to bottom in this direction in Figure 16. The bonds observed in Figure 8b were basically in the *xy* plane, and in the (100) direction the upper and lower halves of a regular hexagon alternately stacked. Unlike the (001) plane loading condition, the bond structure and atomic arrangement of the specimen did not change significantly until the stroke reached 0.06 nm in Figure 17.

When the stroke reached 0.06 nm, the atoms directly under the probe were deformed due to the load exceeding the capacity of the lattice. When we adjusted the observation angle to directly above the loading surface (Figure 17b), we did not find much displacement in the horizontal direction of the germanium atoms whose relative position changed. Therefore, the (110) surface deformed first under the load. As the probe squeezed the atom directly below, it moved downward. During the subsequent 0.09 nm stroke, the atoms directly below the second layer of atoms were again recognized as a complete lattice and were deleted by the software, proving that the subsurface lattice once again overcame the load to return to a normal lattice structure. The surface atom directly below remained at the original position in the *xy* plane until the 0.15 nm stroke in Figure 18, and it broke the atomic bond on one side during the 0.16 nm stroke in Figure 19. At this time, the downward expansion movement of the deformed layer began. When the stroke reached 0.23 nm in Figure 20, the subsurface layer was again recognized as a complete diamond structure. However, the high-shift atom at this time could no longer form a bond with the original atom. At the same time, it can be seen that the atom still connected underwent a position change in the *xy* plane due to the pulling effect brought about by the bond connection and the atomic displacement.

In the process of continuous loading, the specimen loaded on the (110) surface showed a subsurface deformation layer shape different from that of the specimen loaded on the (001) surface. The curvature at the bottom of the deformed layer became very small. The so-called sharp corner shape of the lower surface was pressed to a deeper position inside the monocrystalline germanium model. In contrast, the horizontal extension of the deformed layer was very limited compared to the (001) surface, and the horizontal circumference of the probe loading area was dragged down by the load to a lower extent compared to when the loading surface was (001).

Atomic transient state is shown in Figure 21 at 1.66 nm stroke. When loaded to 3 nm in Figure 22, the curvature of the bottom surface of the deformation layer was similar to the topological shape when the loading depth was 1.66 nm. At the edge of the loaded area, the degree of atomic deformation was lower compared to the center. The potential energy was always much lower inside the deformed layer than around the deformed layer.

Next, we analyzed the (110) crystal plane loading test according to the research method of the deformed layer in the (001) crystal plane loading test.

Figure 23 presents the deformation layer depth data of the (110) crystal plane load test. The thickness of the deformation layer was still judged on the basis of the vertical distance between the tip atom of the probe and the bottom incomplete lattice atom. At the beginning of the loading stage, the deformation layer of (110) fluctuated continuously in the interval of 0–0.9 nm. It can be considered that the monocrystalline germanium specimen was in the elastic deformation stage when the loading first started. After 0.6 nm was loaded, plastic deformation occurred inside the crystal. During the loading process, the crystals continuously “advanced suddenly” and “retreated abruptly”, accompanied by violent fluctuations of the deformation layer of the specimen. The mechanics of the specimen resisted deformation when the (110) surface was loaded.

In the latter part of the loading phase, the fluctuation phenomenon after the rush was no longer obvious. We also performed a regression analysis of the thickness of the deformed layer for the (110) load test. The function’s form was a third-order polynomial equation, and the regression curve is shown as the red line in Figure 23.

#### 3.2.3. Analysis of the Atomic Transient Diagram Loaded on the (111) Crystal Plane

Observing the (111) loading model from the loading direction, it can be seen that the atoms in the middle also had a more obvious layered structure in Figure 24a, and the surface was seeking bonds between nearby atoms to release free energy due to the incomplete crystal lattice. In order to better observe the atomic bond structure of the model, we rotated the monocrystalline germanium model horizontally. When the *x*-axis and our viewing direction formed a 60° angle, the atoms formed a hexagonal grid, and the boundary of the atoms lacked a top atom, preventing them from forming a complete lattice and becoming surface atoms with free energy. Most of them formed bonds with nearby surface atoms and became pentagons.

At the beginning of the loading test, unlike other loading surface tests, the crystal lattice of the (111) loading test did not change significantly until the loading stroke reached 0.39 nm, at which point the atomic bonds were not destroyed (Figure 25). Through the observation of the crystal lattice, we found that, although the bond surface density of the (111) crystal plane model observed directly above was the highest, it was not a triangular bond formed between adjacent layers. Most cases were hexagons, as marked in Figure 24c. In this experiment, the position of the probe tip was exactly at the center of the hexagon, and the first atom where the atomic bond was broken was at the midpoint of the hexagon. The atom squeezed the atom below under the action of the vertical load, and the bond angle between the atom directly below and the surrounding atoms changed.

After that, the atoms under the probe moved in the vertical direction while continuously pushing the horizontal atoms in the opposite direction, which led to the entire deformed layer enveloping the deformed core in Figure 26. The coordinates in the horizontal direction of the atom did not change significantly, indicating that the *z*-coordinate of the atom changed at this stage.

We observed the shape of the deformed layer throughout the loading process and found that the bottom of the deformed layer always maintained a relatively flat shape, and only sporadic vertical crystal lattices would change in advance, laying the foundation for subsequent phase transitions. During the loading process of the (111) crystal plane, the recovery ability of the crystal lattice was only large at the initial stage of deformation. It can be seen that the monocrystalline germanium had a strong shape when the (111) plane was loaded.

The thickness of the deformed layer of the (111) crystal plane loading test was also investigated, and the linear regression of the deformation layer is shown in Figure 27.

Due to reasons previously discussed, the tip of the (111) surface traveled extensively when loaded before actually coming into contact with the atoms of the specimen. When the probe stroke reached 0.5 nm, the surface of the specimen suddenly underwent downward lattice deformation, and the depth of the deformed layer exhibited regular fluctuations. The lattice deformation suddenly expanded to a depth after the load reached a point. The inside of the deformed layer continued to resist deformation and expand in the horizontal direction until the series of changes could not continue to carry the increased load, at which point it again propagated downward.

## 4. Discussion

First, let us discuss the difference in the subsurface deformation of the three crystal plane loading tests. After observing and analyzing the simulation tests of the three crystal planes under vertical load, it can be seen from the results that the change in the monocrystalline germanium loading surface occurred as a function of the topological shape of the bottom of the deformed layer, the thickness of the deformed layer, and other characteristics.

A comparison of the deformation layer thickness data of the three crystal planes is shown in Figure 28. At the beginning of loading, the subsurface of the (001) plane and the (110) plane first reacted to the vertical load, and then all three sides experienced load for a period of time. The fluctuation period of the thickness of the low-deformation layer steadily increased. The thickness of the deformed layer on the (110) plane increased the fastest, and it maintained a higher thickness growth momentum at the end of loading. In addition, we noticed that, for different loading surfaces, the thickness fluctuations of the deformation layer inside the monocrystalline germanium crystal showed obvious differences. The thickness fluctuation of the (001) plane was small when the plane was loaded, while that of the (110) plane differed from that pf the (111) plane. As mentioned earlier, the deformation layer underwent elastic deformation and rebound of the crystal, before a final transformation of elastic deformation into plastic deformation. Therefore, thickness fluctuation represents that the crystal resisted the shape changes upon loading, revealing an ability to deform and restore the lattice state. In order to quantitatively analyze this attribute, we extracted the data from the regression processing of the three curves.

Origin outputs a total of three parameters when performing polynomial regression: the residual sum of squares (RSS), *R*-squared, and adjusted *R*-squared. The residual sum of squares enables calculating the degree of difference between the variable points in the graph and the corresponding regression curve. Therefore, we compared the RSS values of the three faces. As shown in Table 1, the (110) face had the highest RSS value. Thus, the ability to resist deformation was the strongest, and the amplitude of (001) surface oscillation was the smallest.

Furthermore, the topological shape of the deformation layer was also very different. The deformation layer of the (110) crystal plane was significantly sharper, whereas the (001) crystal plane and the (111) crystal plane were mainly flat-bottomed; the bottom surface of the (111) crystal plane deformed layer had a smaller radian than that formed by the (001) crystal plane.

We calculated the number of atoms that were not recognized as diamond lattices when the stroke reached 1 nm, 2 nm, and 3 nm in each crystal plane test, and the results are shown in Table 1. Since the model surface atoms of the (110) plane test and the (111) plane test could not form a complete lattice with the surrounding atoms, they reacted with the surrounding atoms due to their surface activity, and they were identified as non-diamond structures. The experiment removed the non-diamond structures in advance when the relaxation was complete. The statistical results show that, although the non-diamond structure had the largest number of atoms when the (110) crystal plane was loaded, the ratio of the number of atoms was far lower than the ratio between the thicknesses of the deformed layers obtained previously. Moreover, the thickness of the deformation layer of the (001) and (111) planes was not much different, but the number of phase transition atoms in the (001) plane was significantly higher than that in the (111) plane.

Because our probe was not a standard shape, we needed to determine the contact area of the probe projection through the geometric relationship of the size. The hardness H and elastic modulus E of the material can be calculated using the following formula:
(1)$$H=\frac{F_{\max}}{A},$$
where *F*_max_ is the maximum load, and *A* is the projected area of the residual indentation area.

Since the projections of the three probes had the same side length, the projection of the probe was an equilateral triangle. Thus, if the area is required, the side length can be obtained. Since the angle between the probe pyramids was 90°, the side length of the projection was $\frac{\sqrt{2}}{2}$ times the length of the contact edge. The loading depth was 3 nm, and the edge length was $\sqrt{3}$ times the depth; hence, the projected area was $\frac{27}{8}\sqrt{3}\approx 5.846\left(nm^{2}\right)$. The calculated nano-hardness of each crystal plane is shown in Table 1.

Next, we analyzed the energy changes of the three crystal planes; after the three crystal planes relaxed, the potential energies were quite different. The potential energy change is shown in Figure 29. The potential energy value of the (110) face was the smallest when the load was reduced. The potential energy change trends of the three crystal faces were very similar, and, after unloading, they still maintained a short period of potential energy decline, before recovering slightly and remaining stable.

The temperature changes of different loading surfaces are shown in Figure 30. It can be seen that the energy fluctuations in the specimen were relatively severe during loading. We extracted the highest temperature of each crystal plane, and we can see that the (110) plane had the highest temperature rise when subjected to a load. 

## 5. Conclusions

(1)It was found that (110) crystal faces were the hardest and had the greatest impact when loaded. Thus, more work was required for the same loading depth. The (001) face featured intermediate hardness between the (111) and (110) faces, but the smallest subsurface thickness could be obtained during machining. The (111) subsurface lattice had the greatest elasticity under load. However, this crystal surface was the hardest and had a lower temperature rise under load than the other crystal surfaces.(2)The atomic structure analysis showed that smaller subsurface defects could be obtained on the (001) and (111) faces, and that the (111) face was the best loading face due to the lower hardness of the (111) face, being conducive to more efficient processing.

## Figures and Tables

**Figure 1 micromachines-13-00441-f001:**
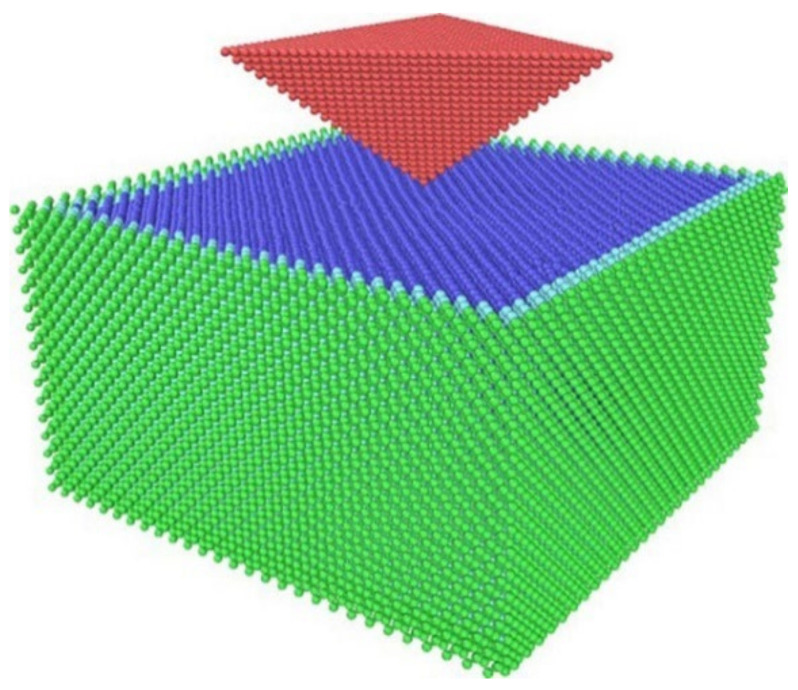
Simulation model of (100) crystal plane nanoindentation.

**Figure 2 micromachines-13-00441-f002:**
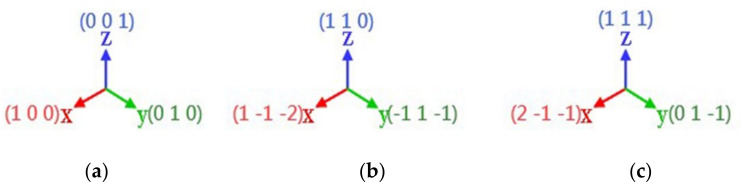
Coordinate axis setting of three crystal plane loading tests. (**a**) (001)coordinate (**b**) (110)coordinate (**c**) (111)coordinate.

**Figure 3 micromachines-13-00441-f003:**
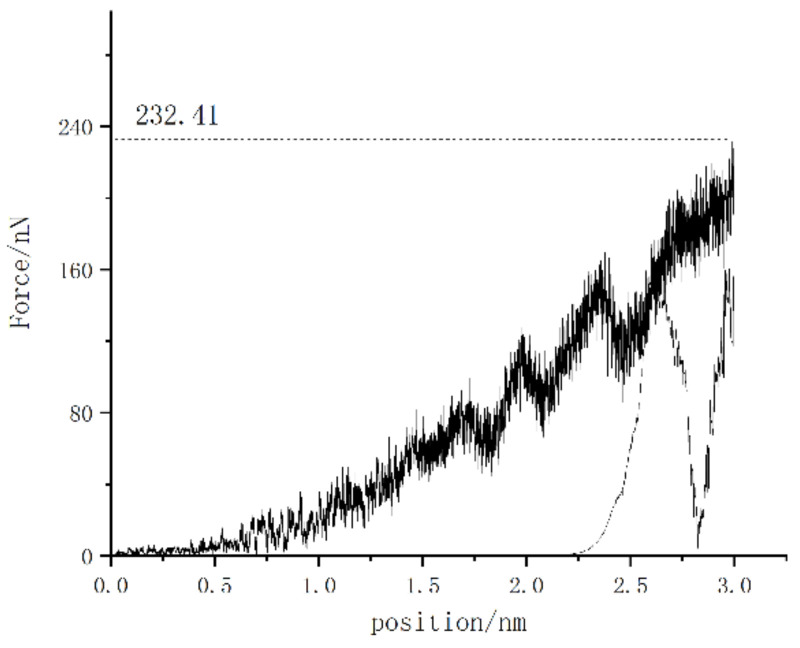
Load–displacement curve of (001) crystal plane loading.

**Figure 4 micromachines-13-00441-f004:**
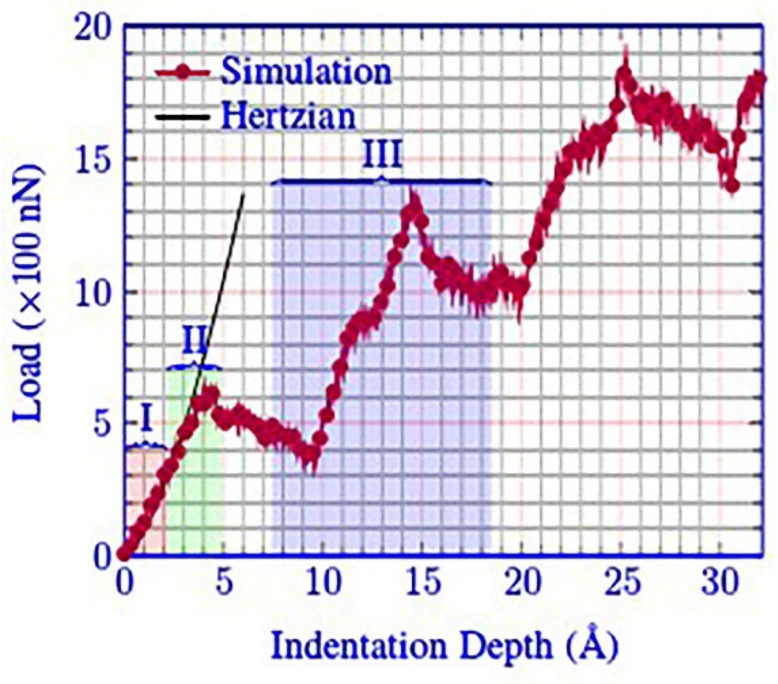
Load–displacement curve of monocrystalline copper.

**Figure 5 micromachines-13-00441-f005:**
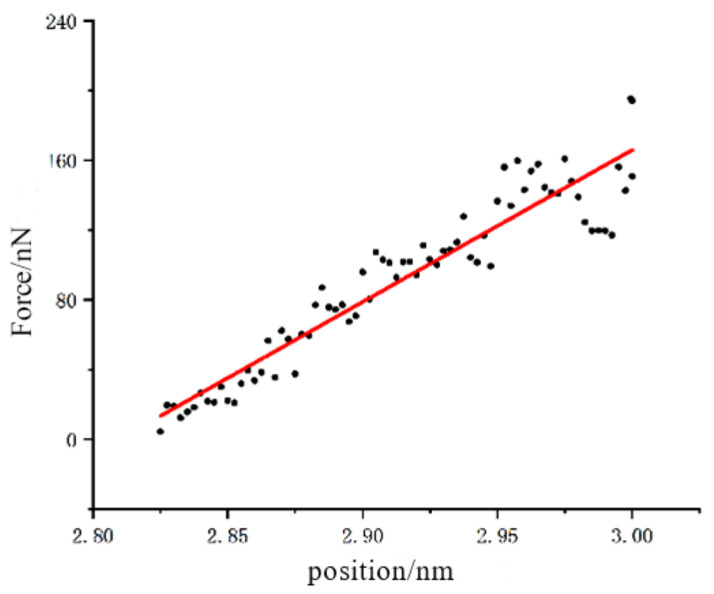
Linear regression of unloaded part of (001) crystal plane test.

**Figure 6 micromachines-13-00441-f006:**
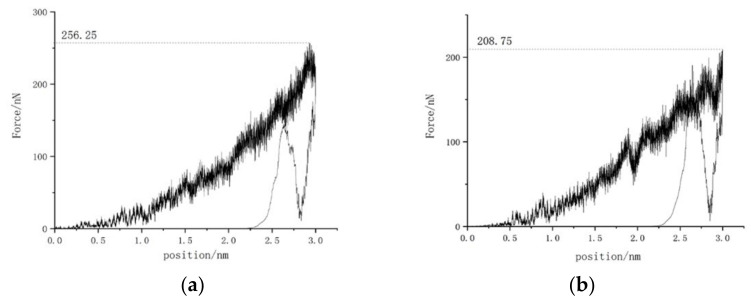
Load–displacement curves of (110) plane and (111) plane. (**a**) (110); (**b**) (111).

**Figure 7 micromachines-13-00441-f007:**
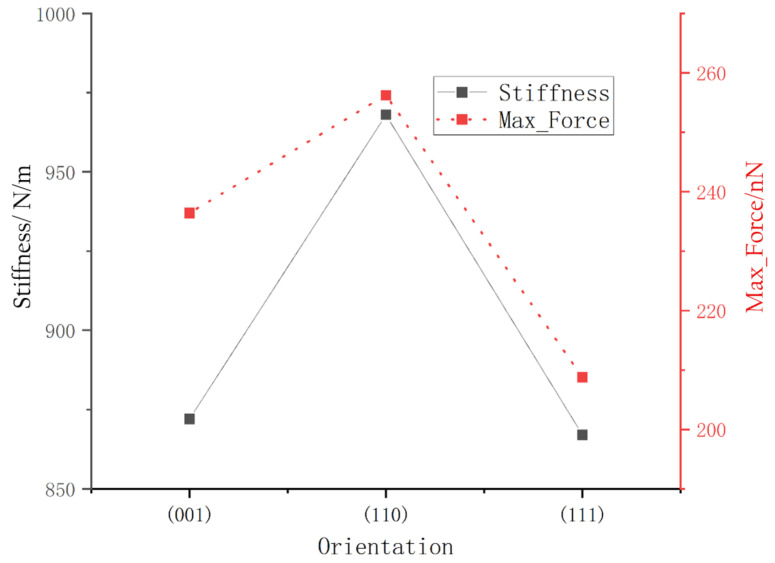
The maximum load and rigidity of the three crystal planes.

**Figure 8 micromachines-13-00441-f008:**

The key structure in the direction of the loaded surface. (**a**) (001) surface; (**b**) (110) surface; (**c**) (111) surface.

**Figure 9 micromachines-13-00441-f009:**
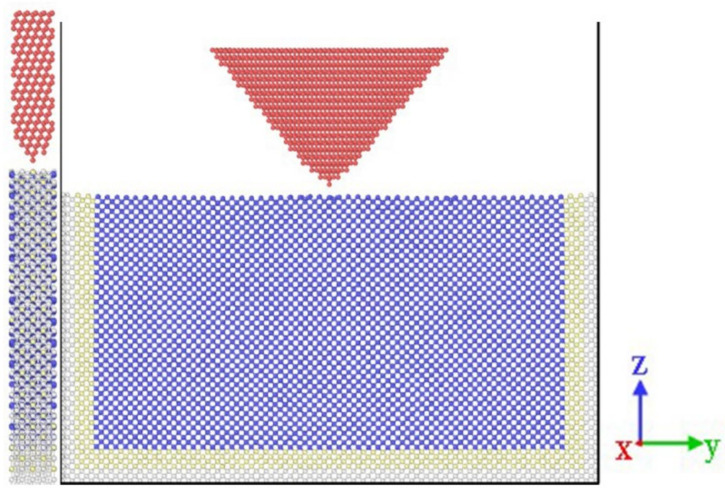
Nanoindentation model slice.

**Figure 10 micromachines-13-00441-f010:**
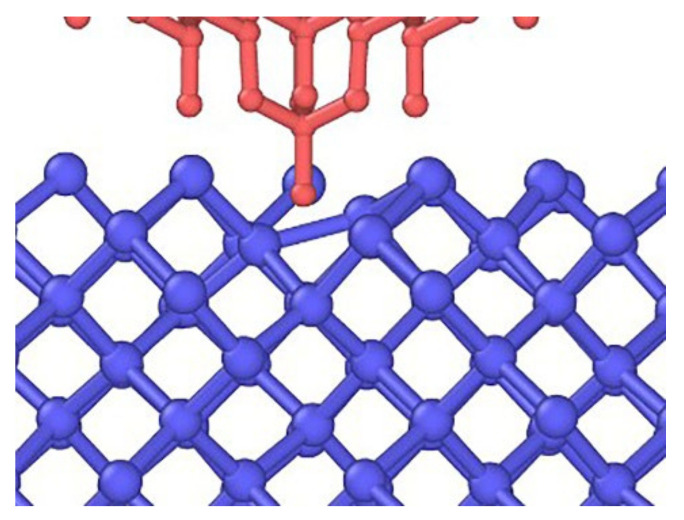
Bond structure when the stroke reaches 0.02 nm.

**Figure 11 micromachines-13-00441-f011:**
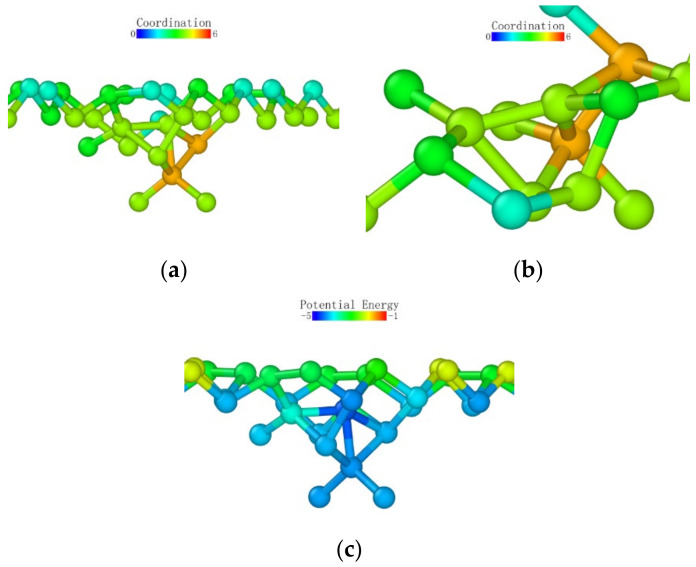
The bond structure when the stroke reached 0.4 nm: (**a**,**b**) colored according to the coordination number; (**c**) potential energy.

**Figure 12 micromachines-13-00441-f012:**
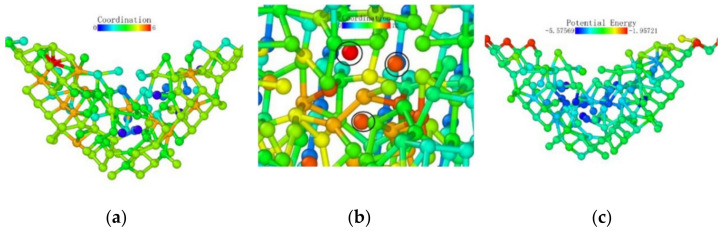
Distribution of coordination number and potential energy at a stroke of 1 nm: (**a**) cutoff radius 2.6 Å; (**b**) cutoff radius 3.2 Å; (**c**) potential energy distribution.

**Figure 13 micromachines-13-00441-f013:**
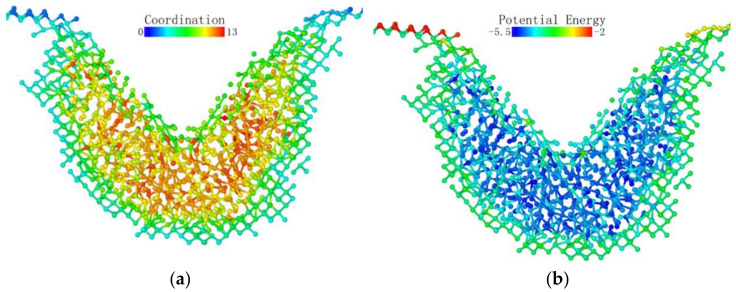
Atomic coordination number distribution (cutoff radius 3.2) and potential energy distribution at a stroke of 3 nm. (**a**) coordination number distribution (**b**) potential energy.

**Figure 14 micromachines-13-00441-f014:**
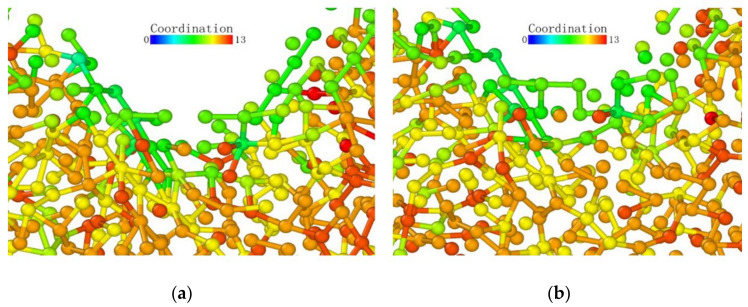
The coordination number at the end of loading (380 ps) and the coordination number at the rebound pole (385.2 ps). (**a**) 380 ps (**b**) 385.2 ps.

**Figure 15 micromachines-13-00441-f015:**
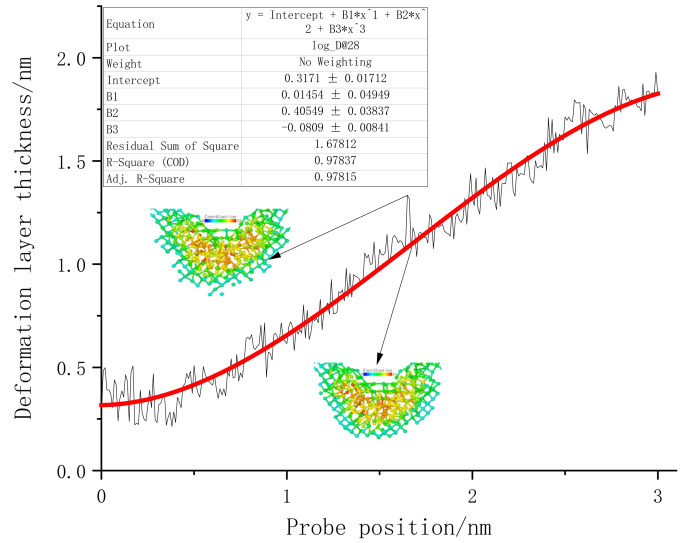
The relationship between the deformation layer and the depth of loading when the (001) surface was loaded.

**Figure 16 micromachines-13-00441-f016:**
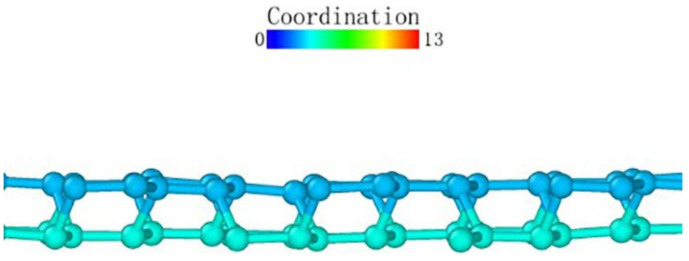
The upper boundary atomic structure before loading.

**Figure 17 micromachines-13-00441-f017:**
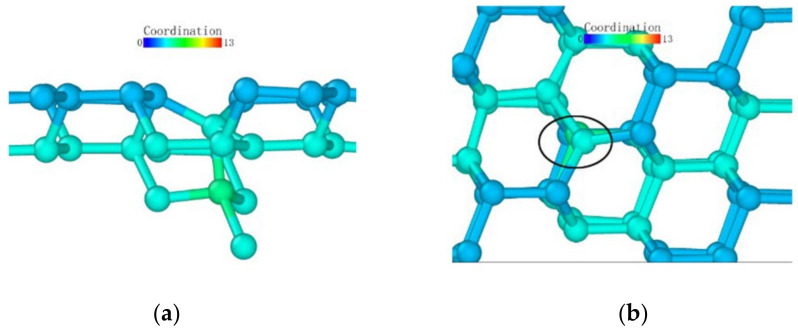
Transient state of the atom directly under the probe when the stroke was 0.06 nm. (**a**) side view (**b**) vertical view.

**Figure 18 micromachines-13-00441-f018:**
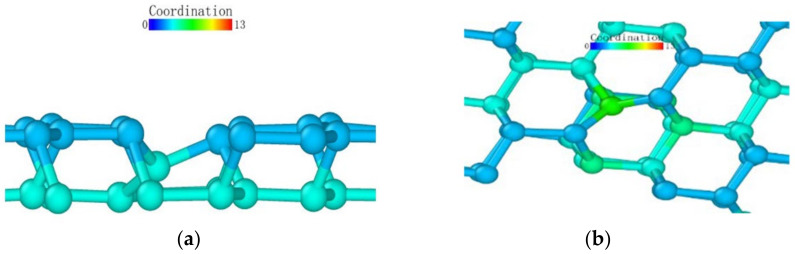
Atomic transient state at 0.15 nm stroke. (**a**) side view, (**b**) vertical view.

**Figure 19 micromachines-13-00441-f019:**
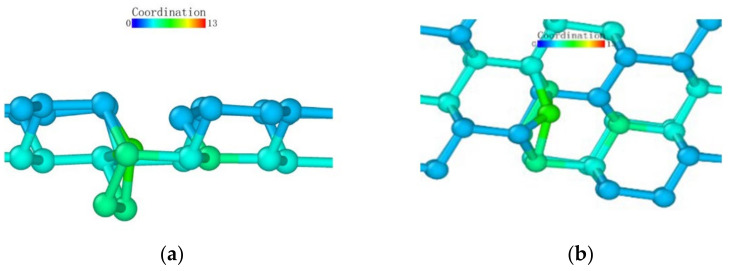
Atomic transient state at 0.16 nm stroke. (**a**) side view, (**b**) vertical view.

**Figure 20 micromachines-13-00441-f020:**
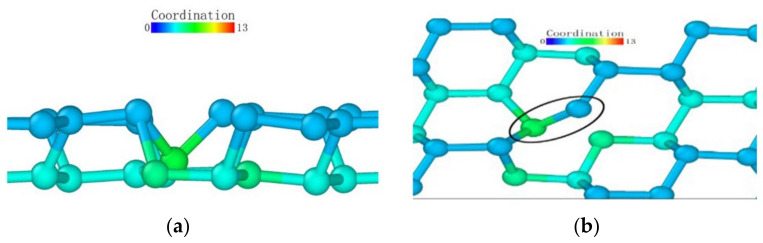
Atomic transient state at 0.23 nm stroke. (**a**) side view, (**b**) vertical view.

**Figure 21 micromachines-13-00441-f021:**
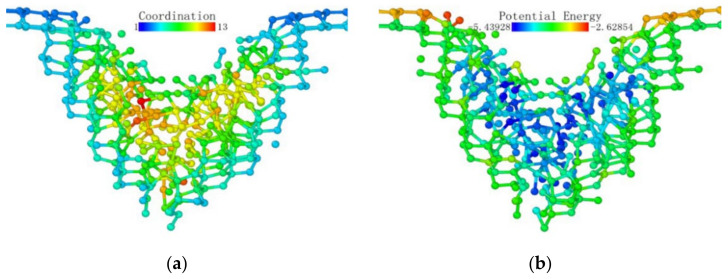
Atomic transient state at 1.66 nm stroke. (**a**) coordination number distribution, (**b**) potential energy.

**Figure 22 micromachines-13-00441-f022:**
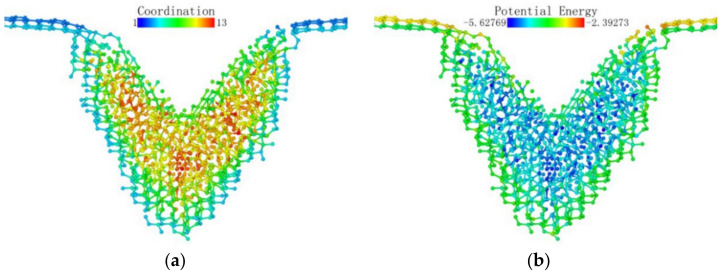
Atomic transient state at a stroke of 3 nm. (**a**) coordination number distribution, (**b**) potential energy.

**Figure 23 micromachines-13-00441-f023:**
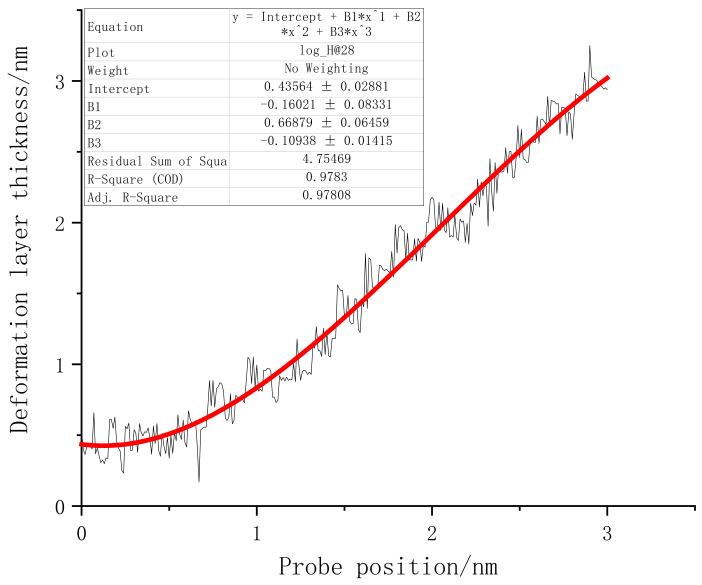
The relationship between the deformation layer and the loading depth when the (110) surface was loaded.

**Figure 24 micromachines-13-00441-f024:**
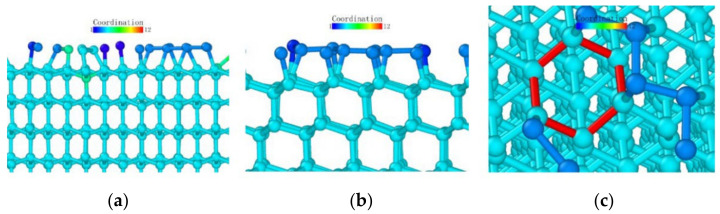
The (111) load model after relaxation. (**a**) side view, (**b**) enlarged side view, (**c**) marked hexagon.

**Figure 25 micromachines-13-00441-f025:**
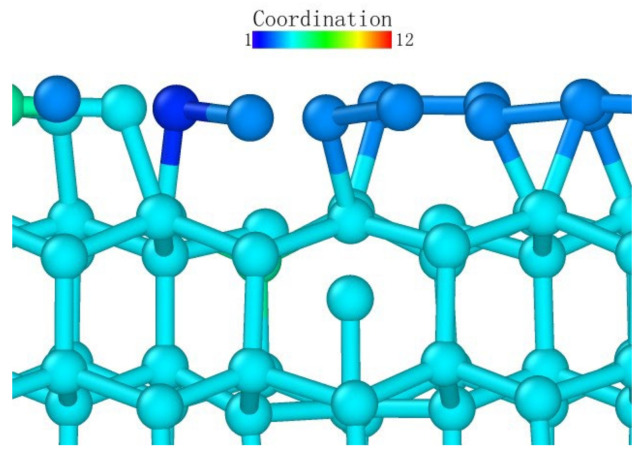
Atomic transient state at 0.39 nm loading stroke.

**Figure 26 micromachines-13-00441-f026:**
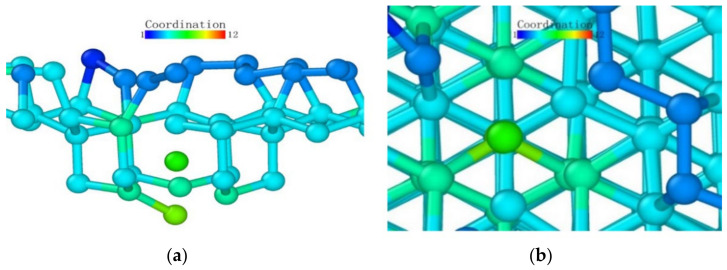
Atomic transient state at 0.48 nm loading stroke. (**a**) side view, (**b**) vertical view.

**Figure 27 micromachines-13-00441-f027:**
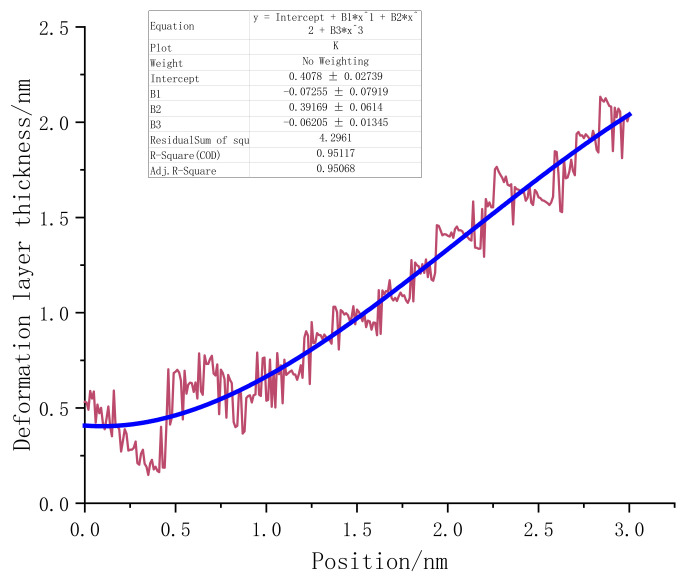
The relationship between the deformation layer and the loading depth when the (111) surface was loaded.

**Figure 28 micromachines-13-00441-f028:**
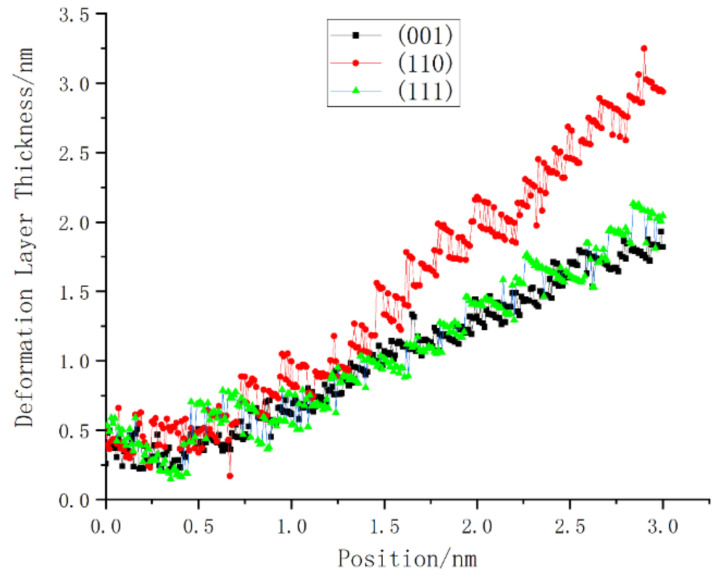
The thickness change of the deformed layer under load on three sides.

**Figure 29 micromachines-13-00441-f029:**
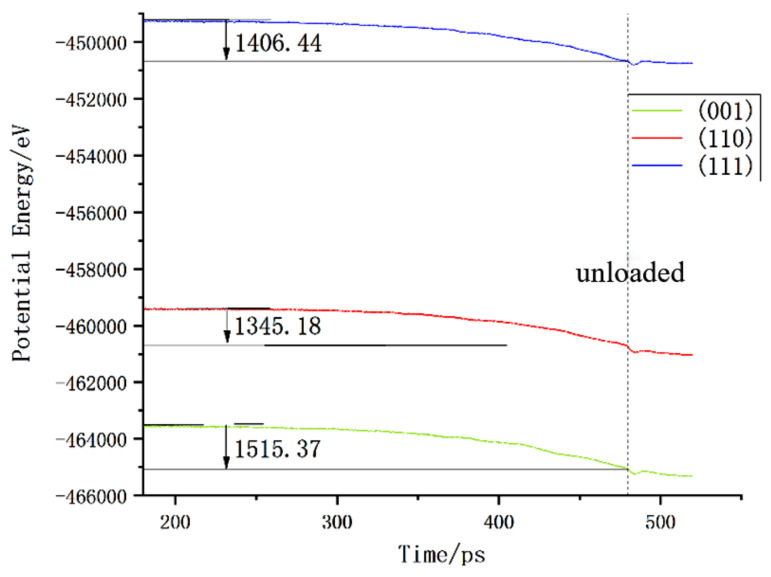
Potential energy changes of different loading surfaces.

**Figure 30 micromachines-13-00441-f030:**
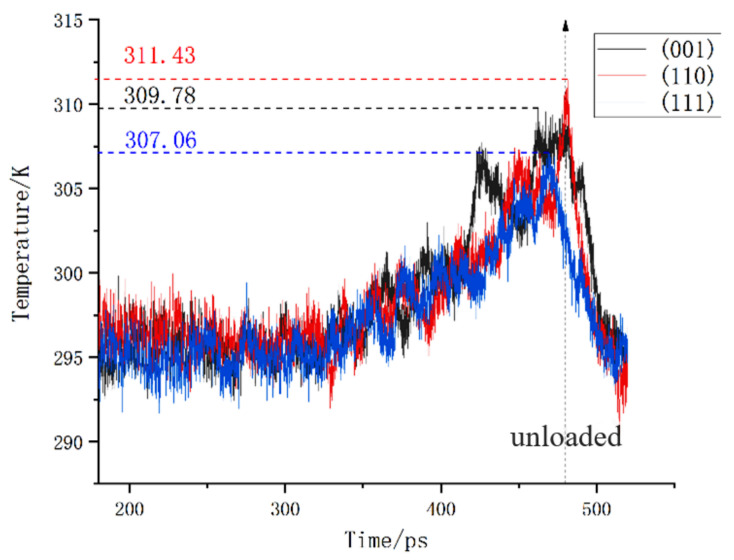
Temperature changes of different loading surfaces.

**Table 1 micromachines-13-00441-t001:** Deformation layer data of crystal plane loading test.

	Attributes	RSS	Deformation Layer Atom Number			Hardness (GPa)
Planes						
			1 nm	2 nm	3 nm	

(001)		1.67812	82	724	2695	38.83
(110)		4.75469	69	727	2753	40.712
(111)		4.2961	117	691	2426	35.58

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
