# Peer review of "Study on the Mechanical Properties of Monocrystalline Germanium Crystal Planes Based on Molecular Dynamics"

_micromachines, 2022, doi:10.3390/mi13030441_

Round 1
Reviewer 1 Report
The work is devoted to molecular dynamics modeling of the process of nanoindentation of germanium crystals. It is shown that under mechanical action the (111) plane is the most optimal for precision machining.
The work can be published after making the appropriate corrections.
1. Lots of broken sentences. This greatly complicates understanding and violates the logic of the presentation of the material (especially see the annotation)
2. Check the presence of dimensions along the axes of the figures
3. The purpose of the work should be clearly stated at the end of section 1.
4. Germanium is a very popular object for research, it is necessary to add data on its mechanical properties in the Introduction, both by nanoindentation and microindentation. There is not a single word in the Introduction about the mechanical characteristics of this crystal.
5. Section 2.1. a very primitive description of the crystal structure, it is necessary to add crystallographic data.
6. Describe the data in detail according to the figure 1. What are these layers?
7. Figures 2 and 3 can be combined, add an axis on the right.
8. Fig. 4 - the value "finally stabilize at Within 0.005 eV/nm" is not obvious.
9. Line 202 "The slope of the linear regression is about 872, that is, the stiffness is 872N/m." where did stiffness come from?
10. Fig.9 there is no description of the second (thin) curves in the figure.
Reviewer 2 Report
The manuscript studied the mechanical properties of germanium using nanoindentation. Overall, I found a few new findings from this research but also a lot of mistakes, including language and figures. In my option, the manuscript cannot be published in Micromachines. Also, I suggested authors revise their manuscript carefully before submitting it to another journal.
The reviewer has some concerns that need to be addressed as follows.
- In the ABSTRACT section, the main findings and innovations need to be clearly stated. Please further polish it and revise the errors, for instance, “the deformable layer thickness, compared cubic The mechanical characteristics…”
- On page 4, what is ‘a’? the lattice constant? Please describe the model as clearly as possible. Also, the nanoindentation model is a widely used loading method, thus it is not necessary to waste a lot of space to introduce this model. Fig.2 to fig.4 can be deleted.
- In Fig. 32, what does the “uninstall” mean? Also, the resolution of all the figures with curves and dots is too low to see clearly, please remake them. Besides the figures, there are too many language errors that keep me from continuing to review.
- In the conclusion part, most of the findings are not different from other nanoindentation simulations.
Round 2
Reviewer 1 Report
Most of the comments were taken into account by the authors of the manuscript.
There are many typos in the text, it is necessary to check the manuscript.
In the Introduction section, there is a lack of actual known data on the mechanical properties of germanium crystals, these data must be added, as indicated in the previous review.
Reviewer 2 Report
There are still too many errors and ambiguities in the article, and I recommend rejecting the manuscript.